# Valorisation of *Ribes nigrum* L. Pomace, an Agri-Food By-Product to Design a New Cosmetic Active

**Hortense Plainfossé [1], Manon Trinel [2], Grégory Verger-Dubois [1], Stéphane Azoulay [2], Pauline Burger [1] and Xavier Fernandez [1,2,*]**

[1]  NissActive, Pépinière InnovaGrasse, Espace Jacques-Louis Lions, 4 traverse Dupont, 06130 Grasse, France; hortense.plainfosse@nissactive.com (H.P.); contact@nissactive.com (G.V.-D.); pauline.burger@nissactive.com (P.B.)

[2]  Université Côte d'Azur, CNRS, ICN, Parc Valrose, CEDEX 2, 06108 Nice, France; Manon.TRINEL@univ-cotedazur.fr (M.T.); Stephane.AZOULAY@univ-cotedazur.fr (S.A.)

*   Correspondence: Xavier.FERNANDEZ@univ-cotedazur.fr; Tel.: +33-4-89-15-01-36

**Abstract:** The ethical and ecological concerns of today's consumers looking for both sustainable and efficient ingredients in finished products, put a lot of pressure on the cosmetic market actors who are being driven to profoundly modify the strategies adopted to innovate in terms of actives while notably being urged to switch from petroleum- to plant-based ingredients. To produce such natural cosmetic ingredients, agri-food by-products are advocated as raw material due to their reduced carbon footprint as they actively contribute to the worldwide improvement of waste management. The process to transform plant waste materials into such powerful and objectified "green" cosmetic actives in compliance with circular economy principles is a long-term integrated process. Such a development is thoroughly exemplified in the present paper through the description of the design of liquid anti-age ingredients based on *Ribes nigrum* L. extract. This was obtained by maceration of blackcurrant pomace. and the embodiment of this extract following its phytochemical analysis notably by HPLC-DAD-ELSD and its bioguided fractionation using in vitro bioassays.

**Keywords:** blackcurrant pomace; *Ribes nigrum* L.; Grossulariaceae; agri-food by-products; sustainability; circular economy; cosmetic ingredient; cosmetic bioassays

## 1. Introduction

Food waste management is a far-reaching topic of concern for our modern society worldwide for environmental and economic reasons, as a substantial amount of food ends up as waste at every stage along the food value chain. Until recently, such waste usually ended up as animal feed, as composting material, in incinerators or in landfills, where it decomposes and generates methane, a potent greenhouse gas [1–3]. However, the generalised environmental concern has triggered both the authorities and industrials to come up with new waste management solutions as stricter regulation has been progressively implemented, notably in the EU (e.g., restricted use of such waste as animal feed, etc.) [2]. The awareness of the marketable potency of such waste greatly contributed to the interest aroused a few years ago and led to innovative waste management solutions considering the circular economy principals. Recycling highly environmentally impacting waste from the food processing industry into raw material to design functional value-added ingredients dedicated to several mainstream sectors of application including nutraceutical and cosmetic sectors appears to constitute a sustainable management solution. Among such initiatives to transform leftovers into new products, one can cite the example of viniculture [4–6]. The global wine industry is flourishing but produces quite a lot of waste: roughly a quarter of the grapes (seeds, stalks, and skins) does not end up

in a bottle. According to the FAO (Food and Agriculture Organization), this industry produces about 14 million tons of pomace every year. Typically discarded in landfills, grape waste is now recognised as detrimental to the environment, causing soil acidification, pest attraction, as well as surface and ground water pollution due to pesticide and fertilizer leaching. Successful valorisation of grape by-products to produce antioxidants, grape oils. and dietary fibres for dietary supplements, pharmaceuticals, etc., not only increases the economic value of grape, but also minimizes its environmental impact [4,5,7].

Developing innovative cosmetic actives, notably anti-aging ingredients, from such agri-food by- products constitutes another potential valorisation channel. Skin, the most voluminous organ of the body, is inevitably submitted to ageing, a complex mechanism driven by both intrinsic (natural chronological aging, genetic, etc.) and extrinsic factors (exposure to environmental factors, including ultraviolet light, weather changes, and both atmospheric and digital pollutions) [8,9]. Reactive Oxygen Species (ROS) are constantly generated by normal cellular processes, but environmental stresses, and UV irradiation may lead to an increased generation of ROS, hence contributing to skin aging [10]. In fact, these species may act as strong oxidizing agents or free radicals, and may activate enzymes (among others, hyaluronidase, collagenase, and elastase) that degrade specifically structural cutaneous building blocks responsible for cutaneous moisturization, e.g., hyaluronic acids, or elasticity and strength, namely elastin and collagen, respectively [8]. According to the World Health Organisation (WHO), the world's population aged over 60 years is expected to total 2 billion by 2050 [11], implying that the anti-aging cosmetic segment still has its best days ahead. The authors hence undertook a survey on the potential revalorisation of several agri-food by-products as anti-aging ingredients while meeting the consumer's demands for sustainability, naturality, transparency, and traceability [12]. A total of 30 extracts was hence obtained by maceration of agri-food by-products in several solvents. The free radical scavenging and anti-inflammatory activities, as well as specific enzyme inhibitory activities of these extracts were assessed in vitro to identify those which efficiently might be used to slow down the skin aging process. The promising antioxidant, anti-hyaluronidase and anti-inflammatory bioactivities evidenced for a blackcurrant pomace hydroalcoholic extract warrant further investigation for the revalorisation and potential use of this agri-food by-product as reliable raw material to design efficient cosmetic actives.

Blackcurrant (*R. nigrum*) is a medium-sized winter-hardy woody shrub growing up to 1.5 m belonging to the Grossulariaceae family that is grown for its small dark purple berries [13–15]. Native to temperate areas of central and northern Europe and northern Asia, the blackcurrant is widely cultivated commercially and domestically across temperate Europe, Russia, New Zealand, parts of Asia and to a lesser extent North America [13,16]. The edible tasteful berries, particularly rich in vitamin C and polyphenols, grow up to 1 cm in diameter and are industrially exploited in the juice market, but also to produce alcoholic beverages, jams, jellies, candies, syrups, and colourings [14]. The blackcurrant annual production averages around 160,000 tons in Europe and 185,000 tons globally [17]. Blackcurrant pomace, also called marc or press-cake, consists of remaining skins, pulp, seeds (estimated to represent roughly 55% of dried blackcurrant pomace [18]), and stems of berries after the fruit has been pressed to make blackcurrant concentrate. It can also sometimes contain some wooden residues and leaf fragments [19]. The processing of fruit berries for juice (no matter the species) generates approximately 20–30% by-products [19], so enormous quantities of blackcurrant pomace are left over at the end of the industrial transformation processes and constitute a manufacturing waste that the agri-food sector must manage [3]. Potentially still a rich source of polyphenols and fibres (including lignin, hemicellulose, cellulose, and pectin [20]), some attempts to exploit the nutritional and economic values of blackcurrant pomace have been made to turn it into food for human consumption, and notably into cereal products (bread, biscuits, etc.) [21]. This article presents the up-grading procedure to design efficient anti-age ingredients based on blackcurrant pomace, while increasing the economic value of the blackcurrant industry, and minimizing its environmental impact.

## 2. Materials and Methods

All chemicals were obtained from Sigma-Aldrich (Saint-Louis, USA) unless otherwise stated.

### *2.1. Blackcurrant Pomace*

Blackcurrant pomace was obtained from an agri-food industry producing fruit juice. The raw material consists of a frozen press cake obtained from berry pressing of organic fruits (varieties: Black Down and Noir de Bourgogne) originating from France.

### *2.2. Blackcurrant Pomace Extraction*

#### 2.2.1. Solid–Liquid Extraction

Blackcurrant pomace was slowly defrosted at room temperature just before extraction to avoid rapid microbial spoilage that would inevitably occur under ambient conditions due to its sugar and water (estimated to reach up to 50% of the pomace mass) contents [3,19,22].

The extractions of agri-food by-products including blackcurrant pomace intended for the general bioactivities screening were performed by direct maceration in an ethanol/water 1/1 (*v/v*) mixture (pomace/solvent ratio 1/5, *w/w*) at room temperature (RT) using a magnetic stirrer (500 rpm) for 2 h. The resulting extracts were then filtered over filter paper 8–12 µm before their bioactivity in vitro assessment.

Then, to assess the best extraction parameters, ethanolic and hydroalcoholic extracts obtained by direct maceration in various ethanol/water mixtures (EtOH/$H_2O$ 100/0, 80/20, 50/50, 20/80 *v/v*; pomace/solvent ratio 1/5, *w/w*) at room temperature using a magnetic stirrer (500 rpm) for 2 h were compared in terms of bioactivities.

In order to develop liquid cosmetic ingredients, extractions were performed by direct maceration of blackcurrant pomace in either glycerine (GreenCoast, Carros, France), propylene glycol (AMI ingredients, Tauxigny, France) or sunflower seed oil (Actibio, Changé, France). About 1 g of plant material was extracted with approximately 10 g of solvent (extraction ratio 1/10, *w/w*) at room temperature using a magnetic stirrer (500 rpm) for 7 h, before filtration over filter paper 8–12 µm.

#### 2.2.2. Ultrasonic-Assisted Maceration

The ultrasonic assistance of maceration was evaluated as follows: approx. 5 g of blackcurrant pomace was placed in a 100 mL beaker and maceration solvent (25 mL EtOH/$H_2O$ 50/50, 50 mL propylene glycol, 50 mL glycerine, and 50 mL sunflower seed oil) was added. The assembly was placed in an ultrasonic bath (3 L, REUS, Contes, France) and the maceration experiments were carried out at room temperature under the following conditions: ultrasound frequency, 25 kHz; time, 20 and 60 min.

### *2.3. Fractionation of R. Nigrum Pomace Extract*

To identify the families of active molecules, an EtOH/$H_2O$ 50/50 extract of *R. nigrum* pomace was then fractionated over C18-silica gel (reverse phase). The fractionation of the bulk extract led to the recovery of 7 distinct fractions: F1 (100 mL water), F2 (100 mL methanol/water 20/80), F3 (100 mL methanol/water 40/60), F4 (100 mL methanol/water 60/40), F5 (100 mL methanol/water 80/20), F6 (100 mL methanol) and F7 (100 mL acetone). The resulting fractions were further evaluated for their bioactivities, and their respective compositions were addressed by HPLC (High Performance Liquid Chromatography).

### *2.4. Bioassays*

The bioassays were performed as presented previously [12,23] in untreated 96-well plates (Thermo Nunc, Villebon-sur-Yvette, France), apart for the lipoxygenase assay which was performed in UV-transparent ones (Costar-Merck, Darmstadt, Germany). Plates were sealed during incubation,

using adhesive films (Greiner Bio-One, Courtaboeuf, France). Samples (extracts, fractions, standards, and controls) were prepared at a concentration of 3.433 mg/mL in dimethyl sulfoxide (DMSO) in 1.5 mL Eppendorf tubes, appropriate for the use of the automated pipetting system epMotion® 5075 (Eppendorf, Montesson, France). The activities of each sample were assessed in triplicate. A microplate reader (Spectramax Plus 384, Molecular Devices, Wokingham, UK) was used to measure absorbance values. Data were acquired with the SoftMaxPro (Molecular devices, Wokingham, UK) software and the Prism software (GraphPad Software, La Jolla, USA) was used to calculate inhibition percentages.

The results are presented as inhibition percentages (I%) calculated as follows:

$$\text{I\%} = [(\text{OD}_{\text{control}} - \text{OD}_{\text{sample}})/\text{OD}_{\text{control}}] \times 100 \text{ (for DPPH, lipoxygenase, elastase and tyrosinase assays)} \tag{1}$$

or as follows:

$$\text{I\%} = [\text{OD}_{\text{sample}}/(\text{OD}_{\text{blank}} - \text{OD}_{\text{control}})] \times 100 \text{ (for hyaluronidase and collagenase assays)} \tag{2}$$

with OD standing for optical density, $\text{OD}_{\text{sample}} = \text{OD}_2 - \text{OD}_1$ ($\text{OD}_1$ and $\text{OD}_2$ being defined for each assay in the following paragraphs), $\text{OD}_{\text{control}}$ corresponding to the DMSO absorbance and $\text{OD}_{\text{blank}}$ to the buffer solution absorbance.

Similarly, all OD values (apart from the hyaluronidase ones) were corrected with the blank measurement corresponding to the optical density of the sample before addition of the substrate.

A positive control consisting in an active commercial cosmetic ingredient specifically selected depending on the assay was tested alongside our samples in strictly the same experimental conditions to perform direct comparison (Table 1). DMSO tested alone constitutes the negative control ($\text{OD}_{\text{control}}$) in each plate: it appears to have no activity on its own.

**Table 1.** Positive controls used in each bioassay performed.

| Assay | Positive Control |
|---|---|
| DPPH Assay | *Rosmarinus officinalis* L. commercial extract (INCI: ROSMARINUS OFFICINALIS EXTRACT; 3.433 mg/mL in DMSO) |
| Tyrosinase Assay | *Phenylethyl resorcinol* (4-(1-phenylethyl)benzene-1,3-diol; INCI: PHENYLETHYL RESORCINOL; 3.433 mg/mL in DMSO) |
| Lipoxygenase Assay | *Arnica montana* L. commercial extract (INCI: ARNICA MONTANA EXTRACT; 3.433 mg/mL in DMSO) |
| Elastase Assay | *Rubus idaeus* L. commercial extract (INCI: RUBUS IDAEUS EXTRACT; 3.433 mg/mL in DMSO) |
| Hyaluronidase Assay | *Rubus idaeus* L. commercial extract (INCI: RUBUS IDAEUS EXTRACT; 3.433 mg/mL in DMSO) |
| Collagenase Assay | Betulinic acid (3.433 mg/mL in DMSO) |

### 2.4.1. DPPH Radical Scavenging Assay

The antioxidant activity of extracts was evaluated based on the scavenging activity of the stable DPPH (1,1-diphenyl-2-picrylhydrazyl) radical [24]: 150 µL of a solution of ethanol/acetate buffer 0.1 M (50/50) was distributed in each well, together with 7.5 µL of the samples evaluated. A first OD reading was performed at 517 nm ($\text{OD}_1$). Then, 100 µL of a DPPH solution (386.25 µM in ethanol) was distributed in each well. The sealed plate was incubated in the dark at RT for 30 min, before performing the $\text{OD}_2$ reading to assess the percentage of inhibition following Equation (1).

### 2.4.2. Tyrosinase Assays

Tyrosinase is a copper-containing enzyme that plays a key role in melanogenesis; it is mainly involved in the hydroxylation of L-tyrosine into L-DOPA and its further oxidation to dopaquinone [25]: 150 µL of a solution of mushroom tyrosinase (171.66 U/mL in phosphate buffer) was distributed in

each well, together with 7.5 μL of the samples evaluated. The filmed plate was incubated at RT for 20 min, before performing the $OD_1$ reading at 480 nm. Then, 100 μL of a solution of substrate (either L-tyrosine or L-DOPA, 1 mM in phosphate buffer) was distributed in each well, and the $OD_2$ reading was performed after 20 min-incubation to assess the percentage of inhibition following Equation (1).

### 2.4.3. Lipoxygenase Assay

Lipoxygenase is an iron-containing enzyme known to play a key role in inflammation [26]: 150 μL of a solution of soybean lipoxygenase (686.66 U/mL in phosphate buffer) was distributed in each well, together with 7.5 μL of the samples evaluated. The filmed plate was incubated in the dark for 10 min. Then, 100 μL of a solution of linoleic acid in phosphate buffer was distributed in each well. After incubation for 2 min in the dark, the $OD_1$ reading was performed at 235 nm; the $OD_2$ reading was performed to assess the percentage of inhibition after a further 50 min-incubation following Equation (1).

### 2.4.4. Elastase Assay

Elastase is an endopeptidase that preferentially digests elastin, the highly elastic protein responsible for the cutaneous firmness, together with collagen [27]. The assays were performed as follows: 150 μL of a solution of porcine pancreatic elastase (0.171 U/mL in Tris buffer) was distributed in each well, together with 7.5 μL of the samples. The filmed plate was incubated at RT for 20 min. $OD_1$ reading was performed at 410 nm, before addition of 100 μL of a solution of *N*-succinyl-Ala-Ala-Ala-*p*-nitroanilide (2.06 mM in Tris buffer). The $OD_2$ reading was performed after 40 min-incubation to assess the percentage of enzymatic inhibition following Equation (1).

### 2.4.5. Hyaluronidase Assay

Hyaluronidases are enzymes that degrade hyaluronic acids which are widely distributed in the body, hence playing a major role in skin aging [28]. The assays were performed as follows: 150 μL of a solution of hyaluronidase (13.3 U/mL in hyaluronidase buffer) was distributed in each well, together with 7.5 μL of the samples evaluated. The filmed plate was incubated at 37 °C for 20 min, and the $OD_1$ reading was performed at 405 nm. Then, 100 μL of a solution of hyaluronic acid (150 μg/mL in pH 5.35 buffer) was distributed in each well. After 30 min incubation at 37 °C, 50 μL of cetyltrimethylammonium bromide (CTAB; 40 mM in a 2% NaOH solution) was added and the $OD_2$ reading was performed to assess the percentage of enzymatic inhibition following Equation (2).

### 2.4.6. Collagenase Assay

Collagenases constitute a family of enzymes that cleave collagen and that are more generally involved in the degradation of the extracellular matrix components, thus leading to sagging skin [29]. Collagenase assays were performed by distributing 150 μL of a solution of collagenase (53 U/mL in tricine buffer) in each well, together with 7.5 μL of the samples. The filmed plate was incubated at RT for 15 min, before the $OD_1$ reading was performed at 345 nm. Then, 100 μL of a solution of 2-furanacryloyl-ʟ-leucylglycyl-ʟ-prolyl-ʟ-alanine (FALGPA; 5.15 mM in tricine buffer) was distributed in each well; the $OD_2$ reading was performed after a 30 min-incubation to assess the percentage of inhibition following Equation (2).

### 2.5. High Performance Liquid Chromatography

Crude extracts and fractions diluted at 10 mg/mL in methanol (MeOH; chromatography grade) and filtered over 0.45 μm PTFE (polytetrafluoroethylene) syringe filter, were analysed using an HPLC Agilent 1200 system (Courtaboeuf, France) equipped with a DAD (Diode Array Detector) and an ELSD (Evaporative Light Scattering Detector) operating under the following conditions: injection volume:

20 μL, and flow rate: 1.0 mL/min. Separations were performed on a C18 column (Phenomenex, Le Pecq, Ile-de-France, France; Luna® 5 μm, 150 mm × 4.6 mm i.d.).

The mobile phase used to analyse the extracts consisted of a multistep gradient of chromatography grade water (A) and acetonitrile (B), both acidified with 0.1% acid formic, and 2-propanol (C): 0–4 min, 2% B; 4–15 min, 2–98% B; 15–20 min, 98% B; 20–25 min, 0–98% C; 25–30 min, 98% C; 30–32 min, 0–98% A.

The DAD was set at 280 nm, and ELSD conditions were set as follows: nebulizer gas pressure 3.7 bar, evaporative tube temperature 40 °C and gain 4.

## *2.6. High Performance Thin-Layer Chromatography*

High performance thin-layer chromatography (HPTLC) was performed on 10 cm × 20 cm HPTLC silica gel 60 F254 pre-coated plates (Merck, Darmstadt, Germany). Standards and samples were applied as 8 mm bands, 12 mm from the left edge and 10.3 or 9.2 mm apart (whether 18 or 20 samples are deposited on the plate) by means of an automated ATS4 sampler (Camag, Muttenz, Switzerland). Two microliters of each standard and samples were applied on the plate. The solvent systems and revelation reagents were chosen according to phytochemicals analysed. The separation on the plate was performed in an automatic ADC2 developing chamber (Camag, Muttenz, Switzerland; developing distance: 7 cm) with the tank previously saturated with the developing solvent system. After developing and drying (5 min), plates were dipped in the detection reagent using the Camag immersion device, dried in a stream of warm air and immersed in a specific revelation reagent. Bands were visualised under visible light using the TLC Visualizer (Camag, Muttenz, Switzerland). The data analysis was performed with a WinCATS Planar Chromatography Manager software (Camag, Muttenz, Switzerland).

### 2.6.1. Amino Acids

The development system used to identify amino acids consisted in an acetonitrile $ACN/H_2O$ 75/25 (*v/v*) system. The plate was treated with ninhydrin (prepared as recommended by CAMAG by dissolution of 0.6 g of ninhydrin in 190 mL of isopropanol, and further addition of 10 mL of glacial acetic acid) [30] to detect amino acid bands using visible light after plate heating at 120 °C for 8–10 min [31].

### 2.6.2. Sugars

The development system used to identify sugars consisted of an $ACN/H_2O$ 75/25 (*v/v*) system. The plate was treated with orcinol prepared in ethanolic sulphuric acid (250 mg of orcinol were solubilised in 100 mL of $EtOH/H_2SO_4$ 95/5 *v/v*) to detect sugar bands using visible light after plate heating at 120 °C for 5–15 min [32].

## 3. Results and Discussion

### *3.1. Hydroalcoholic Blackcurrant Pomace Extract*

As already stated, a series of agri-food by-products were initially selected based on their intrinsic nature, e.g., waste, and their accessibility (easy recovery after the industrial processing, considerable volumes recovered that allow promising revalorization to be pictured, etc.) to evaluate their possible revalorisation as raw materials for the cosmetic ingredient segment. These agri-food by-products were extracted by conventional maceration directly in a hydroalcoholic mixture of solvent ($EtOH/H_2O$ 50/50) convenient to extract polar compounds such as polyphenols known to remain largely in fruits pomace after juice extraction [5,19,33]. Such a hydroalcoholic solvent also revealed itself appropriate for the extraction of the compounds of cosmetic interest from agricultural by- products [23]. Some thirty extracts were recovered, and their whitening, antioxidant, anti- inflammatory and anti-aging properties were evaluated using in vitro assays, and their bioactivities were directly compared to those of commercially available actives. As already stated, the bioactivities of the hydroalcoholic extract

obtained from a 2 h maceration of blackcurrant pomace in EtOH/H$_2$O 50/50 (pomace/solvent ratio 1/5 *w/w*) with an extraction yield of 3.3 ± 0.2% (extraction performed in triplicates) were evaluated in vitro. It displayed promising, but not surprising antioxidant and anti-hyaluronidase bioactivities (Figure 1), given the rich polyphenolic composition and notably the high anthocyanins and their aglycones content of blackcurrant berries [34–37] and pomace [38]. It also displayed to a lesser extend some interesting whitening properties, anti- inflammatory, and anti-collagenase activities, but no anti-elastase activity.

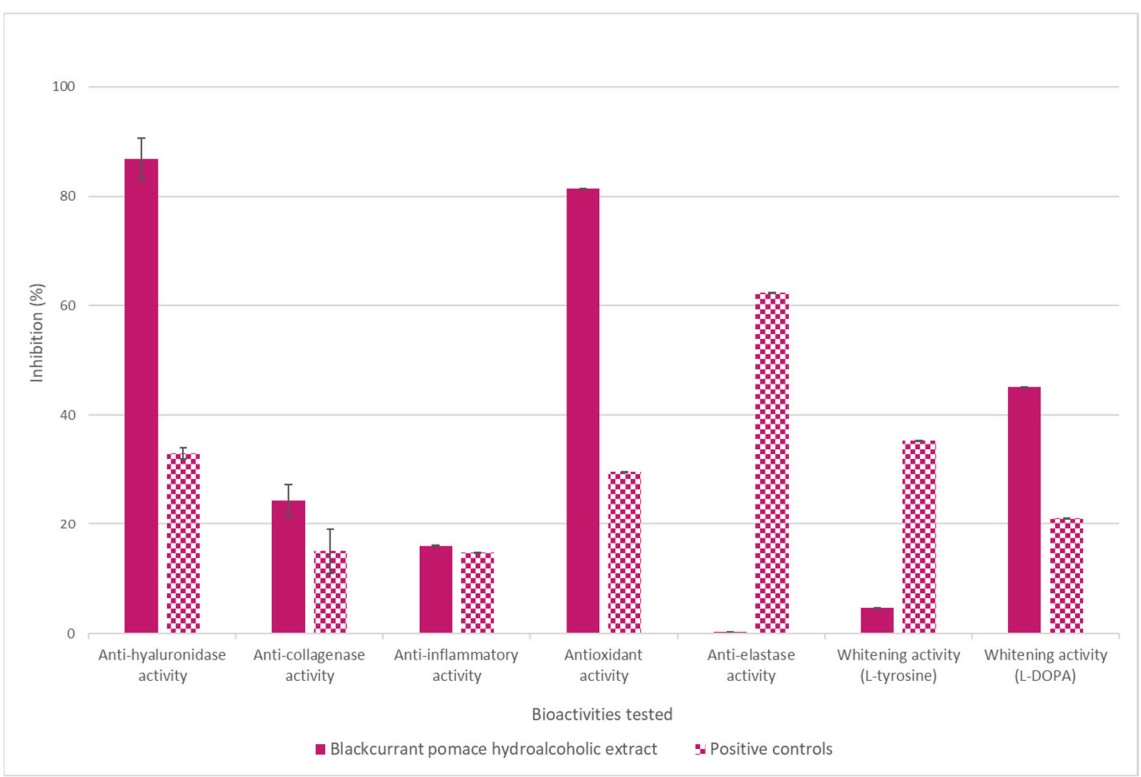

**Figure 1.** Bioactivities of *R. nigrum* pomace extract obtained by direct maceration in EtOH/H$_2$O 50/50 for 2 h at RT, compared to the bioactivities of the positive controls used for the respective activities tested (positive controls).

A larger extract was obtained by maceration of *R. nigrum* pomace in EtOH/H$_2$O 50/50 extract (pomace/solvent ratio 1/5 *w/w*) at RT. The HPLC profile (ELSD chromatogram, Figure 2) revealed this extract's high content in highly polar molecules (group A) displaying retention times inferior to 5 min. The HPLC profile revealed also the presence of less polar molecules (group B) eluting between 5 and 15 min, of which the UV spectra indicate their polyphenolic nature. Group C eluting after 15 min corresponds to a series of terpenes, terpenoids and fatty acids.

High-Performance Thin Layer Chromatography (HPTLC; Figure 3) analyses provided phytochemical information to obtain a quick glimpse of the constituents of group A. Some amino acids have been evidenced using ACN/H$_2$O 75/25 as mobile phase and ninhydrin as reagent [39], but no further characterisation of these amino acids was undertaken given the very low intensities of the corresponding bands. The same mobile phase was used to assess the presence of sugars using orcinol in ethanolic sulphuric acid as derivatisation reagent [32].

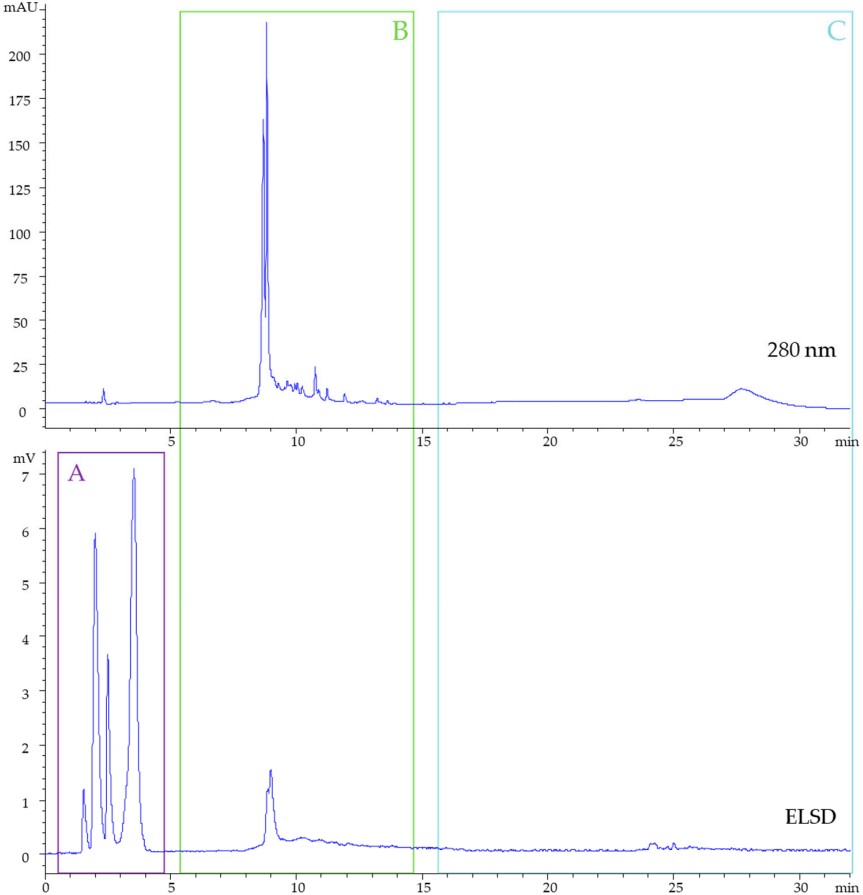

**Figure 2.** HPLC chromatograms obtained on a Luna® C$_{18}$ column (150 × 4.6 mm; 5 μm) at 280 nm and with ELSD, presenting the major families of compounds identified in *R. nigrum* pomace extract (group **A**: polar molecules, group **B**: polyphenols, group **C**: terpenes, terpenoids and fatty acids).

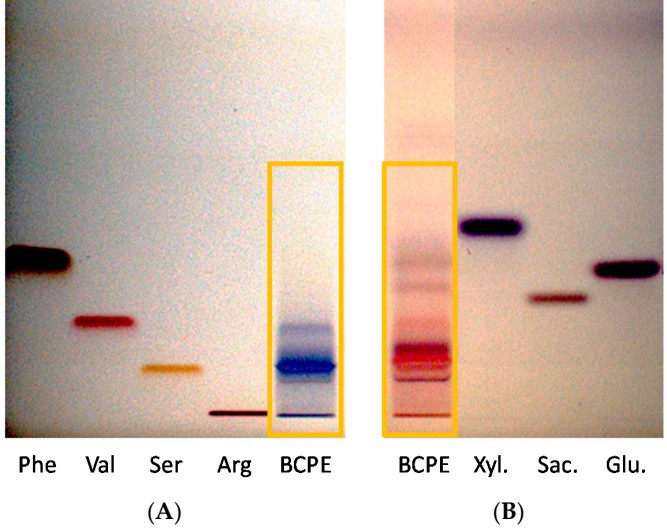

**Figure 3.** HPTLC plate observed under visible light to visualize amino acids (**A**) and sugars (**B**) present in the blackcurrant pomace extract (BCPE) using respectively ninhydrin and orcinol as derivatisation reagents. Tracks: Phe: phenylalanine; Val: valine; Ser: serine; Arg: arginine; Xyl.: xylose; Sac.: saccharose and Glu.: glucose.

To characterize the active compounds responsible for the bioactivities evidenced, this EtOH/H$_2$O 50/50 extract of *R. nigrum* pomace was then fractionated over C18-silica gel (Table 2). The fractionation of 647 mg of blackcurrant led to the recovery of seven distinct fractions: F1 (water), F2 (MeOH/H$_2$O 20/80), F3 (MeOH/H$_2$O 40/60), F4 (MeOH/H$_2$O 60/40), F5 (MeOH/H$_2$O 80/20), F6 (methanol) and F7 (acetone). The resulting fractions were analysed by HPLC (Figure 4) and further evaluated in vitro for their bioactivities the same way as the ones of the crude extract. Results of these bioassays are presented in Table 2.

**Table 2.** Bioactivities of *R. nigrum* pomace extract and of the corresponding fractions F1–F7.

| | | *R. Nigrum* Pomace Extract | F1 | F2 | F3 | F4 | F5 | F6 | F7 |
|---|---|---|---|---|---|---|---|---|---|
| | Mass (mg) | 647.0 | 416.0 | 62.9 | 96.4 | 21.4 | 10.2 | 18.3 | 1.1 |
| | Yield (%) | 100.0 | 64.3 | 9.7 | 14.9 | 3.3 | 1.6 | 2.8 | 0.2 |
| Bioactivities | Anti-hyaluronidase | ++++ | - | ++++ | ++++ | ++++ | ++++ | ++++ | - |
| | Anti-collagenase | +++ | + | ++++ | +++ | ++ | + | - | - |
| | Anti-inflammatory | - | - | +++ | +++ | ++++ | ++ | - | - |
| | Antioxidant | +++ | - | ++++ | ++++ | ++++ | +++ | - | - |
| | Anti-elastase | - | - | - | - | - | - | - | - |
| | Whitening (L-tyrosine) | ++ | - | ++ | ++ | +++ | ++ | - | - |
| | Whitening (L-DOPA) | + | - | + | + | + | ++ | + | - |

(−): inhibition < 30%; (+): 30% < inhibition < 50%; (++): 50% < inhibition < 70%; (+++): 70% < inhibition < 90%; (++++): inhibition > 90%.

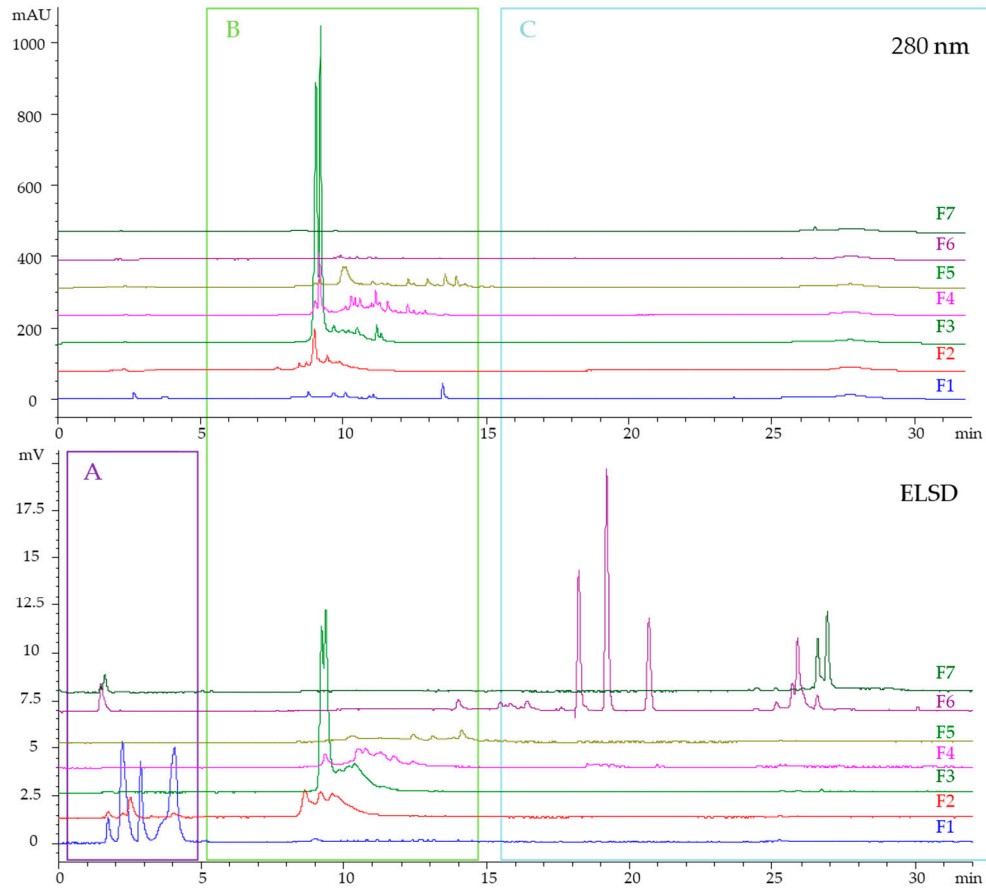

**Figure 4.** HPLC chromatograms obtained on a Luna® C$_{18}$ column (150 × 4.6 mm; 5 µm) at 280 nm and with ELSD for the 7 fractions obtained by fractionation of the *R. nigrum* pomace extract (group **A**: polar molecules, group **B**: polyphenols, group **C**: terpenes, terpenoids and fatty acids).

As already stated, neither the crude extract nor any of the fractions display anti-elastase activity. Fraction F1, obtained with the best yield, and constituted mainly of compounds of group A, e.g., sugars, amino acids, peptides (Figure 4), displays none of the activities evaluated. Similarly, mainly made up of lipophilic compounds from group C, fractions F6 and F7 display no remarkable bioactivity except some good anti-hyaluronidase activity for F6, probably due to the occurrence of several lipophilic (group C) compounds such as terpenes, terpenoids or fatty acids. Polyphenolic molecules of intermediary polarities (group B) were collected in fractions F2–F5, which all display excellent anti-hyaluronidase, anti-inflammatory and antioxidant bioactivities, as well as promising anti-collagenase and whitening potentials. The anti-inflammatory and antioxidant activities of polyphenolic molecules contained in blackcurrant berries, as well as in pomace are already known [14,36,37]. The whitening activity of blackcurrant juice was also already stated in the literature [40], and one can imagine that a certain amount of the compounds responsible for this whitening activity remains in the pomace after the berries' processing. However, the anti-hyaluronidase and anti- collagenase activities of this pomace extract were to our knowledge evidenced for the first time for this pomace extract and together with its anti-inflammatory, antioxidant, and whitening activities, bestowed this extract complete anti-aging activity to valorise in a cosmetic active.

Maceration improvement attempts were then undertaken to obtain a blackcurrant pomace extract concentrating the active molecules in order to develop the most active anti-age ingredient. The efficacy of various solvent mixtures was evaluated, as well as the assistance of ultrasound during maceration (Table 3). The resulting extracts were analysed by HPLC (Figure 5) and their bioactivities were evaluated in vitro as previously.

No significant improvement in terms of extraction yields could be evidenced by modulation of the $H_2O$/EtOH ratio (Table 3). Regarding the bioactivities of the extracts obtained by conventional maceration, the hydroalcoholic extract 50/50 displays the best results, however, this difference cannot be explained by the major discrepancy in the phytochemical profile (Figure 5).

**Table 3.** Bioactivities of various *R. nigrum* pomace extracts obtained by conventional and ultrasound-assisted macerations.

| | | Conventional Maceration | | | | | Ultrasound-Assisted Extraction | |
|---|---|---|---|---|---|---|---|---|
| | Solvent | EtOH | $H_2O$/EtOH 20/80 | $H_2O$/EtOH 50/50 | $H_2O$/EtOH 80/20 | $H_2O$/EtOH 50/50 | $H_2O$/EtOH 50/50 | $H_2O$/EtOH 50/50 |
| | Maceration parameters | RT, 2 h | RT, 2 h | RT, 2 h | RT, 2 h | RT, 6 h | RT, 20 min | RT, 60 min |
| | Extraction yield (%) | 3.6 ± 0.3% | 2.2 ± 0.1% | 3.2 ± 0.2% | 3.2 ± 0.2% | 2.8 ± 0.6% | 3.6% | 3.9% |
| Bioactivities | Anti-hyaluronidase | + | - | ++++ | + | +++ | ++++ | ++++ |
| | Anti-collagenase | - | - | +++ | - | +++ | ++++ | ++++ |
| | Anti-inflammatory | - | - | - | - | - | ++ | + |
| | Antioxidant | ++ | ++ | +++ | +++ | +++ | ++++ | ++++ |
| | Anti-elastase | - | - | - | - | - | - | - |
| | Whitening (L-tyrosine) | + | - | ++ | ++ | + | ++ | ++ |
| | Whitening (L-DOPA) | + | - | + | + | + | + | ++ |

(−): inhibition < 30%; (+): 30% < inhibition < 50%; (++): 50% < inhibition < 70%; (+++): 70% < inhibition < 90%; (++++): inhibition > 90%.

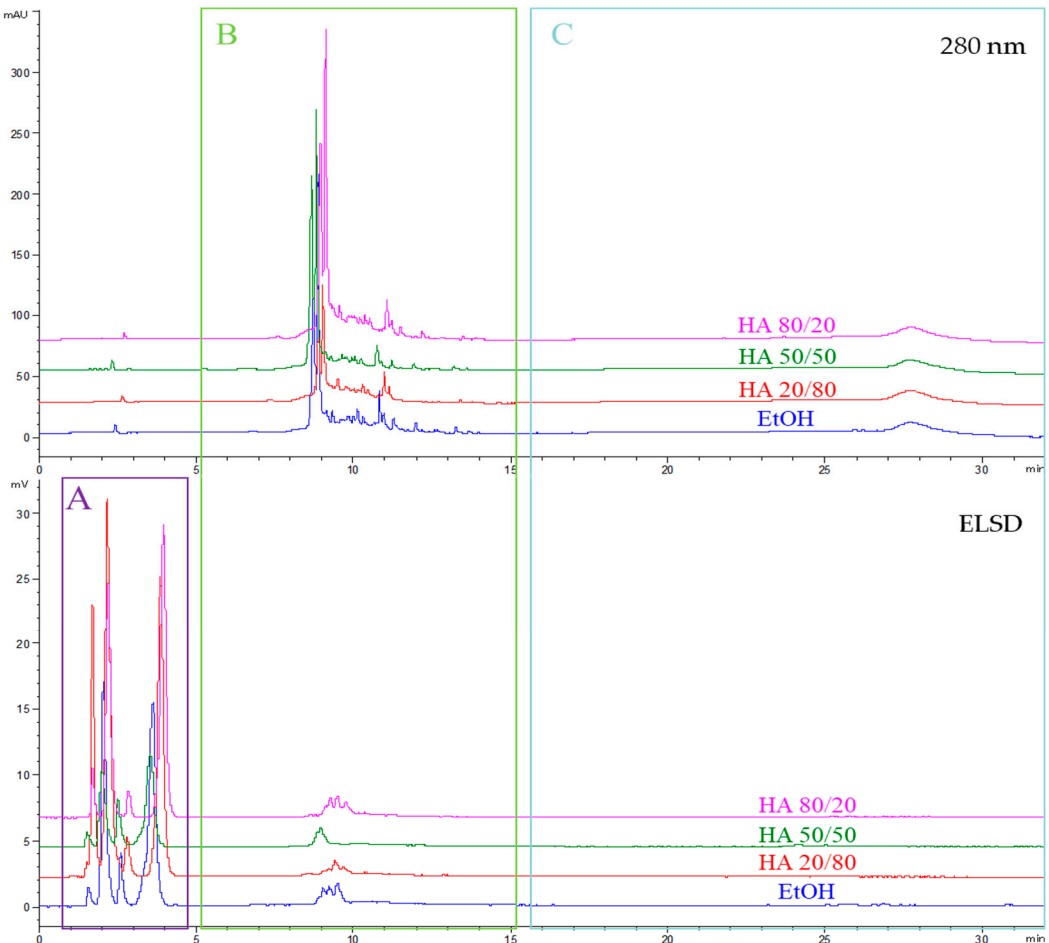

**Figure 5.** HPLC chromatograms obtained on a Luna® C$_{18}$ column (150 × 4.6 mm; 5 μm) at 280 nm and with ELSD for the various *R. nigrum* pomace extracts obtained by conventional maceration (HA: hydroalcoholic, group **A**: polar molecules, group **B**: polyphenols, group **C**: terpenes, terpenoids and fatty acids).

Increasing the maceration duration using this 50/50 ratio to 6 h does not seem to present any advantage: no significant bioactivity improvement could be revealed and does not justify and counterbalance the increased energy consumption necessary to extend the maceration duration.

We hence tested the effect of ultrasonic assistance on such a maceration. UAE (Ultrasound-Assisted Extraction) is based on the mechanical breakdown of a plant matrix as a result of the cavitation phenomenon due to the application of ultrasound. Once the acoustic pressure is high enough, cavities generated by compression and relaxation cycles collapse sparking mechanical forces which lead to the disruption of biomembranes: the content of the plant cell is then released in the solvent [41,42]. The application of ultrasound leads to slightly higher extraction yields (Table 3). As their HPLC chromatograms are similar to the one obtained by maceration, they are not presented in the current article. One can also notice that the bioactivities reported for an extract obtained in only 20 min are equivalent, if not higher, than those of the corresponding conventional extract using H$_2$O/EtOH 50/50. However, increasing the UAE duration to 60 min did not lead to higher bioactivities. This simple and efficient technique, already used for a lot of applications, including the extraction of phytochemicals using water, ethanol, and their mixtures, may constitute a time-saving alternative to conventional techniques to obtain such extracts dedicated to the development of high added values ingredients for nutraceuticals, cosmetics, pharmaceuticals, etc. [43].

However, one should keep in mind that such low extraction yields (< 5%) are not economically viable and would not constitute a rational route of agri-food waste management that could be

implemented at larger scale to efficiently reduce amounts of leftovers generated by this sector. That is why, attempts to develop a liquid cosmetic active were then undertaken and the economic viability of the resulting ingredients was discussed in terms of effective agri-food waste reduction. To do so, the selection of appropriate liquid supports was guided by the physico-chemical properties of the active compounds identified in the hydroalcoholic extract.

## 3.2. Liquid Cosmetic Ingredient

Active plant extracts can usually not directly be integrated into cosmetic formulations mainly due to solubility issues, but also as they may display undesirable colour, offensive odour, inappropriate viscosity, etc. [44,45]. Further processing, including the deposition of the extract on the appropriate cosmetic support to ease incorporation into a cosmetic formulation, may be necessary. The extraction of plant material by hydroalcoholic solvents as presented previously, usually requires time-consuming steps of solvent evaporation and further resolubilisation of the dry extract before its deposition on a cosmetic support, either liquid or solid.

Alternatively, liquid cosmetic ingredients, often preferred by formulators, may be obtained by direct extraction of the plant material in solvents appropriate for their ultimate integration in a finished cosmetic product. In such a case, one can consider the analysis of the chemical composition of the hydroalcoholic extract and its bioguided fraction performed previously as a pre-requisite to identify the appropriate cosmetic support that would offer maximised access to the active molecules in order to obtain the most concentrated, e.g., the most active ingredient.

To extract blackcurrant pomace, propylene glycol and glycerine were selected for their polarities quite similar to the ones of hydroalcoholic mixtures, their water-solubility, and their reduced water activity, bacteriostaticity, and fungistaticity that imply self-preserving properties [46]. Propylene glycol and glycerine are among the most widely used cosmetic supports: they serve as humectant in many personal care formulations including facial cleansers, moisturizers, etc. [23,46].

Sunflower (*Helianthus annuus L.*) seed oil is a non-volatile and non-fragrant plant oil presenting a decent colour that is used in cosmetics as an emollient. It is also appreciated for its smoothing effect towards signs of cutaneous stress or irritation. Easily absorbed by skin, it is a non-comedogenic and highly moisturising oil that is largely use in a variety of personal care products [47]. Sunflower seed oil, even more lipophilic than the hydroalcoholic mixtures previously used, was incorporated in this study to try to access the more lipophilic compounds, notably compounds of group C (which are notably quite abundant in fraction F6 displaying interesting anti-hyaluronidase activity, (Table 2, Figure 4), remaining in blackcurrant pomace and to decipher if, once concentrated in a solvent, they might display some cosmetic interest.

To prepare such extracts, maceration parameters defined earlier for the development of liquid ingredients were used [23]: blackcurrant pomace macerated in either PG, glycerine or sunflower seed oil (pomace/solvent ratio 1/10 *w/w*) under stirring for 7 h at RT. The resulting extracts were then filtered over filter paper, and their activities, as well as the ones of solvents alone (no activity reported; data not shown) were assessed using in vitro bioassays (Table 4).

Unsurprisingly, given the chemistry of pomace extract presented previously, the sunflower seed extract does not display any interesting bioactivity. The propylene glycol and glycerine display some antioxidant and anti-collagenase activities, and almost no anti-hyaluronidase activity. Although not as interesting as the hydroalcoholic extract, the possibility to improve those later extractions was undertaken, increasing notably the maceration duration. As evidenced in Table 4, increasing the extraction duration up to 24 h appears to have almost no effect when using glycerine as extraction solvent. On the other hand, the resulting propylene glycol 24 h-extract displays more interesting anti- hyaluronidase, antioxidant and anti-collagenase activities compared to the propylene glycol 7 h-one. Given those results, propylene glycol was considered as the most appropriate solvent for the development of such blackcurrant pomace liquid ingredient.

**Table 4.** Bioactivities of various *R. nigrum* pomace extracts obtained by conventional and ultrasound-assisted maceration in propylene glycol, glycerine, and sunflower seed oil.

| | Solvent | Sunflower Seed Oil | Propylene Glycol | Glycerine | Propylene Glycol | Glycerine | Propylene Glycol | Glycerine |
|---|---|---|---|---|---|---|---|---|
| | Maceration parameters | RT, 7 h | RT, 7 h | RT, 7 h | RT, 12 h | RT, 12 h | RT, 24 h | RT, 24 h |
| Bioactivities | Anti-hyaluronidase | - | + | - | + | - | +++ | - |
| | Anti-collagenase | - | +++ | +++ | ++ | ++ | ++++ | +++ |
| | Anti-inflammatory | - | - | - | - | - | - | - |
| | Antioxidant | - | ++ | ++ | ++ | ++ | +++ | ++ |
| | Anti-elastase | + | - | - | - | - | - | - |
| | Whitening (L-tyrosine) | - | - | - | - | - | - | - |
| | Whitening (L-DOPA) | - | - | - | - | - | - | - |

(−): inhibition < 30%; (+): 30% < inhibition < 50%; (++): 50% < inhibition < 70%; (+++): 70% < inhibition < 90%; (++++): inhibition > 90%.

The assistance of ultrasound was than tested for 20 and 60 min to enhance the dynamics of the maceration process of blackcurrant pomace in propylene glycol; the bioactivities of the resulting extracts were evaluated as previously and directly compared to those of conventionally obtained extracts (Figure 6). The assistance of ultrasound during the maceration process does not appear to be interesting in this case to enhance the extraction of active molecules. One can consider it as a time-saving technology to obtain in 1 h, an extract displaying similar activities to the one obtained in 7 h by conventional maceration. However, the economic viability of such a technology at an industrial scale for such a gain remains questionable, notably taking into consideration the real bioactivity improvements observed by the simple increase of maceration duration to 24 h.

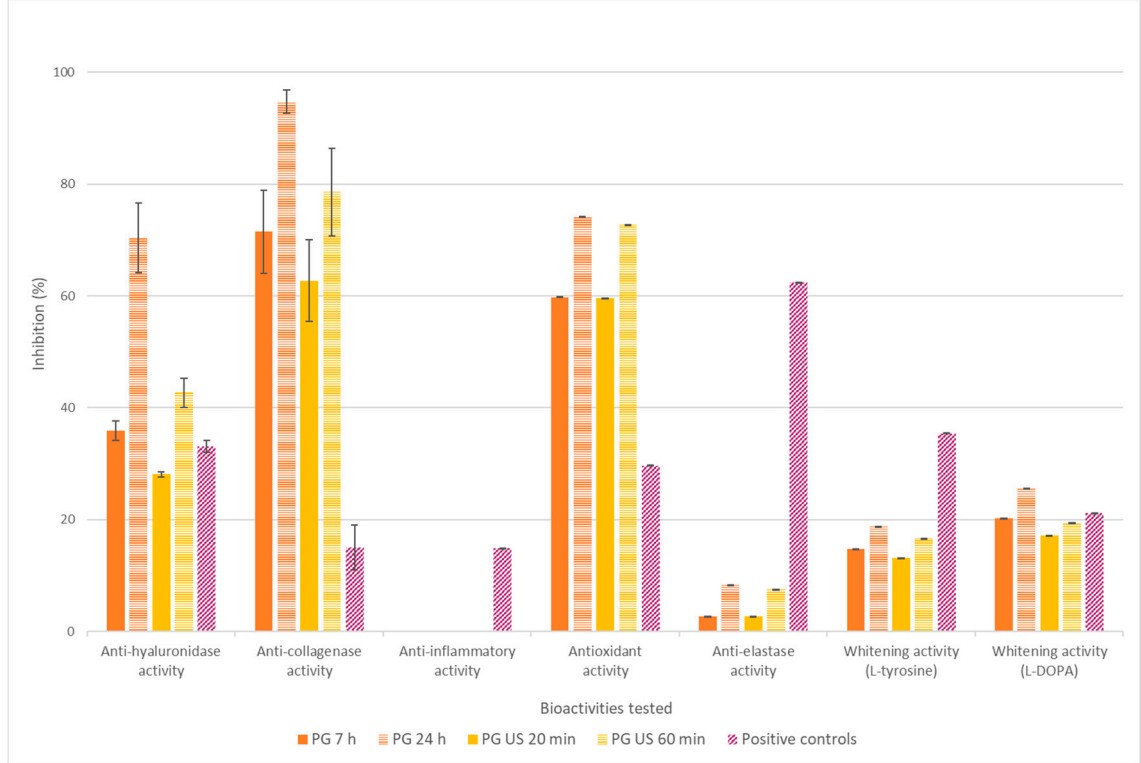

**Figure 6.** Bioactivities of *R. nigrum* pomace extract obtained by ultrasound-assisted maceration for 20 and 60 min, compared to the bioactivities of extracts obtained by conventional 7 h- and 24 h- macerations.

## 4. Conclusions

Driven by consumer concerns for product efficacy and naturality, one can observe an influx on the cosmetic ingredients market, of fruit- and vegetable-based ingredients over the last decade. The revalorisation of agri-food by-products as raw material to develop new cosmetic ingredients hence constitutes an alternative to these regular plant-derived ingredients and an opportunity for brands to stay trendy while ensuring sustainability. In this article, we presented an integrated strategy to develop a such natural anti-aging cosmetic active using blackcurrant pomace, an agri-food by- product as starting raw material. Undoubtedly, such by-products constitute economically attractive sources of raw material for the development of high added-value ingredients that align with the consumers' green ethic. Upcycling such an agri-food by-product solves the raw material sourcing issue, that constitutes the pivotal point in the development of a new natural cosmetic ingredient. As the sourcing already exists, there is no additional delay to the commercialisation of the final ingredient due to the cultivation of the raw material. It also extends the initial raw material's value via the generation of new revenue lines, while reducing the related waste treatment costs and offers an appreciable degree of traceability on the materials' origin. Furthermore, using agricultural by-products can engage cosmetics suppliers and customers through their shared responsibility toward environmental preservation and gain new market shares. However, some questions remain: despite that, are such products made up from waste marketable? Is upcycling really viable in such a marketing-orientated sector as the cosmetic one? Are the average consumers, and not only the environmentally concerned ones, ready to use products made from refuse? Furthermore, such ingredient development processes are not as inexpensive as initially thought. In fact, waste collection requires workforce, waste treatment has a cost, as some noxious compounds that may be present in the by-product after initial processing have to be removed, and finally some specific extraction techniques required for waste transformation may be quite expensive. Fortunately, more and more consumers prioritise sustainability and we can only expect this tendency to expand as recycling is not an option anymore in the current context of industrial growth all over the globe.

**Author Contributions:** Conceptualization, H.P., M.T., G.V.-D., S.A., P.B. and X.F.; methodology, H.P., P.B. and X.F.; validation, H.P., P.B., S.A. and X.F.; formal analysis, H.P. and M.T.; investigation, H.P. and M.T.; resources, G.V.-D. and X.F.; data curation, H.P., M.T., S.A., P.B. and X.F.; writing—original draft preparation, P.B. and H.P.; writing—review and editing, H.P., M.T., G.V.-D., S.A. and X.F.; supervision, X.F.; project administration, G.V.-D. and X.F.; funding acquisition, G.V.-D. and X.F. All authors have read and agreed to the published version of the manuscript.

**Funding:** This research project was financially supported by the French ANRT—Association Nationale de la Recherche et de la Technologie, grant number 0036/2016.

**Acknowledgments:** The authors wish to thank Julien Lesage for kindly giving us access to the blackcurrant by-products.

**Conflicts of Interest:** The authors declare no conflict of interest.

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
