# Peer review of "Valorisation of Ribes nigrum L. Pomace, an Agri-Food By-Product to Design a New Cosmetic Active"

_cosmetics, doi:10.3390/cosmetics7030056_

Round 1

Reviewer 1 Report

In this article, an integrated strategy to develop a natural anti-aging cosmetic ingredience – blackcurrant pomace was presented. The product represents economically attractive sources of raw materials. However, blackcurrants (Ribes nigrum) are relatively new to the U. S. market, but they are well known and popular in Europe and Asia. Numerous scientific studies confirm the beneficial effect on maintining the proper structure and functioning of the skin and its appendages. For this reason, they are important in cosmetology as ingredients for cosmetic preparation.

Abstract

12-21     I recommend modifying the abstract – focusing more on the methods used and the results of the work.

Introduction

30           I recommend not including information from 2010, to update the information or shorten

Material and Methods

76           to create the abbreviation Ribes nigrum

129         to write R. nigrum in italics

166,180,189,197,206       why references are in bold

295         to correct Figure 1 to Figure 3

I recommend unify format of tables – Table 1 and 3 are framed compared to Table 2

Throughout the text, when it comes to cosmetic ingredients, state the names according to INCI.

Conclusion

I recommend reworking the conclusion, not quoting references, using your own ideas based on the results of the work.

References

499         to add information to the source 11?

Author Response

Reviewer 1

Open Review

English language and style

( ) Extensive editing of English language and style required 
( ) Moderate English changes required 
( ) English language and style are fine/minor spell check required 
(x) I don't feel qualified to judge about the English language and style 

Yes

Can be improved

Must be improved

Not applicable

Does the introduction provide sufficient background and include all relevant references?

( )

( )

(x)

( )

Is the research design appropriate?

(x)

( )

( )

( )

Are the methods adequately described?

(x)

( )

( )

( )

Are the results clearly presented?

(x)

( )

( )

( )

Are the conclusions supported by the results?

( )

( )

(x)

( )

Comments and Suggestions for Authors

In this article, an integrated strategy to develop a natural anti-aging cosmetic ingredience – blackcurrant pomace was presented. The product represents economically attractive sources of raw materials. However, blackcurrants (Ribes nigrum) are relatively new to the U. S. market, but they are well known and popular in Europe and Asia. Numerous scientific studies confirm the beneficial effect on maintining the proper structure and functioning of the skin and its appendages. For this reason, they are important in cosmetology as ingredients for cosmetic preparation.

Abstract

12-21     I recommend modifying the abstract – focusing more on the methods used and the results of the work.

The last sentence of the abstract has been modified accordingly.

Introduction

30           I recommend not including information from 2010, to update the information or shorten

The information was removed to shorten, according to the suggestion.

Material and Methods

76           to create the abbreviation Ribes nigrum

The abbreviated version was used

129         to write R. nigrum in italics

This modification has been done.

166,180,189,197,206       why references are in bold

This problem has been addressed

  • 295         to correct Figure 1 to Figure 3 This modification has been done.

I recommend unify format of tables – Table 1 and 3 are framed compared to Table 2 the tables have been harmonized

Throughout the text, when it comes to cosmetic ingredients, state the names according to INCI. The INCI name of the commercial ingredients used as controls have been added in table 1, page 4.

Conclusion

I recommend reworking the conclusion, not quoting references, using your own ideas based on the results of the work.

Some modifications of the conclusion were done according to the suggestion

References

499         to add information to the source 11?

URL of the website was added

Reviewer 2 Report

Manuscript ID: cosmetics-855339

The manuscript entitled “Valorisation of Ribes nigrum L. pomace, an agri-food by-product to design a new cosmetic active” focuses on a very interesting aspect that has been well developed in recent years, and that is related to sustainability concerns. In this article, the process to obtain raw materials from food by-products that normally are discarded as waste, with anti-aging properties, for subsequent use in cosmetic formulations is described in a clear and well-structured manner. Thus, the article should be published after a minor review. Some suggestions for improving the manuscript are described below:

  • Line 55: the word "aging" does not seem to be necessary.
  • Line 63: It would be better "...namely elastin and collagen, respectively".
  • Line 110 and line 115: It would be better "(pomace/solvent ratio 1/5, w/w)".
  • Line 125: replace “…were added” by “…was added”.
  • Line 142: replace “mg/ml” by “mg/mL”.
  • Line 216: Place (polytetrafluoroethylene) after PTFE.
  • Line 244: replace “ml” by “mL”.
  • Figure 1: The resolution of chart legends should be increased.
  • Figure 5: The meaning of “HA” should be explained in the legend.
  • Line 366: replace “...in term of...” by “...in terms of...”.
  • Line 436: replace “...ingredient...” by “...ingredients...”.
  • Line 530: replace “…axtract” by “...extract”.
  • Why ultrasound-assisted maceration chromatograms are not shown?
  • Finally, a deeper discussion of the results would be important.

Author Response

Reviewer 2

Open Review

English language and style

( ) Extensive editing of English language and style required 
( ) Moderate English changes required 
(x) English language and style are fine/minor spell check required 
( ) I don't feel qualified to judge about the English language and style 

Yes

Can be improved

Must be improved

Not applicable

Does the introduction provide sufficient background and include all relevant references?

(x)

( )

( )

( )

Is the research design appropriate?

(x)

( )

( )

( )

Are the methods adequately described?

(x)

( )

( )

( )

Are the results clearly presented?

(x)

( )

( )

( )

Are the conclusions supported by the results?

(x)

( )

( )

( )

Comments and Suggestions for Authors

Manuscript ID: cosmetics-855339

The manuscript entitled “Valorisation of Ribes nigrum L. pomace, an agri-food by-product to design a new cosmetic active” focuses on a very interesting aspect that has been well developed in recent years, and that is related to sustainability concerns. In this article, the process to obtain raw materials from food by-products that normally are discarded as waste, with anti-aging properties, for subsequent use in cosmetic formulations is described in a clear and well-structured manner. Thus, the article should be published after a minor review. Some suggestions for improving the manuscript are described below:

  • Line 55: the word "aging" does not seem to be necessary. This modification has been done.
  • Line 63: It would be better "...namely elastin and collagen, respectively". This modification has been done.
  • Line 110 and line 115: It would be better "(pomace/solvent ratio 1/5, w/w)". These modifications have been done.
  • Line 125: replace “…were added” by “…was added”. This modification has been done.
  • Line 142: replace “mg/ml” by “mg/mL”. This modification has been done.
  • Line 216: Place (polytetrafluoroethylene) after PTFE. This modification has been done.
  • Line 244: replace “ml” by “mL”. This modification has been done.
  • Figure 1: The resolution of chart legends should be increased. This modification has been done.
  • Figure 5: The meaning of “HA” should be explained in the legend. This modification has been done.
  • Line 366: replace “...in term of...” by “...in terms of...”. This modification has been done.
  • Line 436: replace “...ingredient...” by “...ingredients...”. This modification has been done.
  • Line 530: replace “…axtract” by “...extract”. This modification has been done.
  • Why ultrasound-assisted maceration chromatograms are not shown?

A sentence was added Line 357 to explain.

  • Finally, a deeper discussion of the results would be important.

Some modifications of the conclusion were done according to the suggestion
